# An Analysis of JADE2 in Non-Small Cell Lung Cancer (NSCLC)

**DOI:** 10.3390/biomedicines11092576

**Published:** 2023-09-19

**Authors:** Ciara Murphy, Glòria Gornés Pons, Anna Keogh, Lisa Ryan, Lorraine McCarra, Chris Maria Jose, Shagun Kesar, Siobhan Nicholson, Gerard J. Fitzmaurice, Ronan Ryan, Vincent Young, Sinead Cuffe, Stephen P. Finn, Steven G. Gray

**Affiliations:** 1Department of Histopathology, Labmed Directorate, St. James’s Hospital, D08 NHY1 Dublin, Irelandstephen.finn@tcd.ie (S.P.F.); 2Thoracic Oncology Research Group, Central Pathology Laboratory, Trinity St. James’s Cancer Institute (TSJCI), St. James’s Hospital, D08 RX0X Dublin, Irelandakeogh5@tcd.ie (A.K.); 3Faculty of Biology, University of Barcelona, 08025 Barcelona, Spain; 4Department of Histopathology and Morbid Anatomy, Trinity College Dublin, D02 PN40 Dublin, Ireland; 5School of Medicine, Trinity College Dublin, D02 PN40 Dublin, Ireland; 6Surgery, Anaesthesia and Critical Care Directorate, St. James’s Hospital, D08 NHY1 Dublin, Irelandvyoung@stjames.ie (V.Y.); 7HOPE Directorate, St. James’s Hospital, D08 NHY1 Dublin, Ireland; 8Department of Clinical Medicine, Trinity College Dublin, D02 PN40 Dublin, Ireland; 9School of Biological Sciences, Technological University Dublin, D07 XT95 Dublin, Ireland

**Keywords:** non-small cell lung cancer, JADE2, overexpression, prognosis, mutation, tumor mutational burden, tumor-infiltrating immune cells

## Abstract

The JADE family comprises three members encoded by individual genes and roles for these proteins have been identified in chromatin remodeling, cell cycle progression, cell regeneration and the DNA damage response. JADE family members, and in particular JADE2 have not been studied in any great detail in cancer. Using a series of standard biological and bioinformatics approaches we investigated JADE2 expression in surgically resected non-small cell lung cancer (NSCLC) for both mRNA and protein to examine for correlations between JADE2 expression and overall survival. Additional correlations were identified using bioinformatic analyses on multiple online datasets. Our analysis demonstrates that JADE2 expression is significantly altered in NSCLC. High expression of JADE2 is associated with a better 5-year overall survival. Links between JADE2 mRNA expression and a number of mutated genes were identified, and associations between JADE2 expression and tumor mutational burden and immune cell infiltration were explored. Potential new drugs that can target JADE2 were identified. The results of this biomarker-driven study suggest that JADE2 may have potential clinical utility in the diagnosis, prognosis and stratification of patients into various therapeutically targetable options.

## 1. Introduction

The most commonly diagnosed cancer in the world is lung cancer. In 2020, it was estimated that of 19.3 million new cancer cases worldwide, approximately 2.2 million or 11.4% will be lung cancer. In terms of mortality, lung cancer was estimated at 1.8 million deaths representing 18% of all cancer deaths [1]. Lung cancer can be classified into two major subtypes, small cell lung cancer (SCLC) and non-small cell lung cancer (NSCLC), with the vast majority (approximately 85%) falling into the latter subtype. Traditionally treatment of lung cancer has been associated with poor outcomes; however, recent advances in immunotherapy [2], targeted therapy [3] and the discovery of new actionable mutations [4] have greatly increased the treatment options available, in particular for NSCLC.

Aberrant regulatory mechanisms are common in NSCLC and it is now well established that epigenetics is one such regulatory mechanism with significant alterations in NSCLC [5]. Epigenetics can be loosely defined as a series of modifications on either the DNA or histones which together can collectively be called the “epigenetic code” or “epigenome” [6,7]. This code is generated and utilized by a large number of proteins which can be classified as either “readers”, “writers” or “erasers” depending on their function to regulate gene expression [8]. Of these, a large number of proteins contain a plant homeodomain (PHD) that recognizes either methylated or acetylated residues on histones [8,9,10] and are considered “…central “readers” of histone post-translational modifications” [9]. Numbering more than 100 proteins [9], these include a small subset of proteins called the JADE family [11]. Originally identified and described as “Gene for Apoptosis and Differentiated Epithelia” [12], this family comprises three members designated as PHF17 (JADE1), PHF16 (JADE2) and PHF15 (JADE3) [11].

Epigenetics has now been classified as a new “Hallmark of Cancer” with a designation for “nonmutational epigenetic reprogramming” as an enabling characteristic that facilitates the acquisition of original hallmark capabilities, and also enables the provisional new hallmark capability of phenotypic plasticity [13], and as such defining such the functional roles of these epigenetic readers in cancer is of importance. Our knowledge regarding the actual functional roles of the JADE family is limited. The initial characterization of JADE1 found that it formed an interaction with the von Hippel–Lindau (pVHL) protein and the lysine acetyltransferase Tip60 (KAT5) to direct acetylation at histone H4 [14]. Further studies determined that JADE1 is also part of the HBO1 complex, a large complex containing multiple histone/lysine acetyltransferases (including Tip60), and was found to be critical with respect to this complex directing acetylation specifically at histone H4 to regulate transcription [15,16]. Subsequent studies have found that the N-terminal region of JADE1 acts as the platform necessary to bring the catalytic HBO1 subunit to its cognate H3–H4 substrate in this regard to elicit acetylation [17]. The HBO1 complex itself plays important roles in various molecular processes including transcription, DNA replication, and DNA repair [18], and JADE1 has been shown to play essential roles in regulating chromatin during mitosis [19,20]. JADE2 has been shown to form a complex with the lysine methylase LSD1 during neurogenesis and cognitive function [21,22], and has also been linked to myogenesis [23]. The HBO1 complex has a distinctive characteristic in that it has the presence of three PHD finger domains in two different subunits: tumor suppressor proteins ING4/5 and JADE1/2/3. The PHD domains in JADE1 were shown to be essential for (a) HBO1 complexes to bind chromatin and (b) for a critical JADE1-associated tumor suppressor activity [24], and it is important to note that the HBO1 complex has been shown to play important roles in cancer [18].

Despite the large body of evidence linking the HBO1 complex with cancer, overall, there have been very few studies examining the actual role of JADE family proteins themselves in cancer. Critical roles for JADE1 have been identified in renal cell carcinoma [25,26,27,28,29]. In ovarian cancer, JADE2 has been shown to be highly overexpressed and may function to regulate YAP1 a critical ovarian cancer oncogene [30], whilst a gene fusion between JADE2 and NUP98 has been observed in a pediatric acute myeloid leukemia [31]. In colon cancer, JADE3 has been shown to promote increased tumourigenicity by increasing cancer stem cell properties [32].

However, an examination of JADE family members in NSCLC has not been studied to any great degree. Most recently, a link between a single nucleotide polymorphism (SNP—rs329118) in JADE2 and the risk of developing NSCLC has been identified [33]. As such, in this manuscript, we set out to assess the expression of JADE2 in NSCLC to determine if its expression also had any potential utility as a biomarker in lung cancer at both the mRNA and protein levels. We show that JADE2 is predominantly downregulated in NSCLC at the mRNA but not at protein level, and that expression of JADE2 has prognostic value and is associated with a better 5-year overall survival (OS). Links between JADE2 expression and other genes commonly associated with NSCLC are described, along with the identification of novel mutated genes that affect JADE2 expression. Correlations between JADE2 expression and various parameters such as tumor mutational burden and immune cell infiltration are explored. The results presented suggest that JADE2 may be a candidate biomarker in NSCLC and warrants further study.

## 2. Materials and Methods

### 2.1. Nomenclature

Nomenclature for genes and proteins used the current HUGO definitions [34].

### 2.2. Cell Culture

Twelve cell lines (comprising normal human bronchial epithelial cell lines (HBECs) and NSCLC cancer cell lines) were used in this study comprising: normal bronchial epithelial cells (HBEC3, HBEC4 and HBEC5 Beas2B) and NSCLC cell lines with various histotypes as follows: adenocarcinoma (A549, NCI-H2228, NCI-H1975, HCC827 and H3122), squamous cell carcinoma (SKMES-1), large cell carcinoma (NCI-H460) and adenosquamous (NCI-H596).

HBECs 3–5 [35], HCC827, NCI-H1975 and NCI-H3255 were a gift from Prof. John D Minna (Hamon Centre for Therapeutic Oncology Research, UT Southwestern, Dallas, TX, USA). The A549, SKMES-1, NCI-H460, NCI-H1299 and NCI-H596 cell lines were purchased from the ATCC (LGC Promochem, Teddington, UK). NCI-H3122 cells were a kind gift from Dr. Dong-Wan Kim, Seoul National University Hospital, Seoul, Republic of Korea).

All cells were maintained in a humidified atmosphere containing 5% CO_2_ in appropriate media supplemented with 10% fetal bovine serum (FBS), and Antibiotic Antimycotic Solution (Sigma-Aldrich, St. Louis, MO, USA: Cat No: A5955) with the exception of HBECs. HBECs were grown in Keratinocyte Growth Medium 2 (Promocell, Heidelberg, Germany; Cat. No: C-20111), and on collagen-coated plates. A549 cells were grown in Nutrient Mixture F-12 Ham (Sigma-Aldrich, St. Louis, MO, USA—Cat. No: N6658), SKMES-1 in Dulbecco′s Modified Eagle′s Medium—high glucose (Sigma-Aldrich, St. Louis, MO, USA—Cat. No: D6429). All other NSCLC cell lines were grown in RPMI-1640 Medium (Sigma-Aldrich, St. Louis, MO, USA—Cat. No: R8758). All cell lines were routinely tested for mycoplasma as per the published PCR protocol [36].

### 2.3. Primary Tumor Samples

Chemotherapy naïve tumor specimens surgically resected were used in this study. All samples were evaluated by a pathologist immediately following resection and tumor tissue along with matched normal tissue were removed and flash-frozen for downstream analysis. Comprising 11 adenocarcinomas and 11 squamous cell carcinomas a summary of their clinical and histopathological data is summarized in Table 1. Prior to surgery informed consent for bio-banking was obtained from each patient, and for retrospective analyses individual consent was waived. The study proceeded only after formal approval from the SJH/AMNCH Hospital Ethics Committee—Ethics REC (No.: 041018/8804; Project ID: 0624), and in accordance with the Declaration of Helsinki (as revised in 2013).

### 2.4. Formalin Fixed Paraffin Embedded Samples

A total of 204 surgically resected NSCLC tumor specimens from the period 1999–2007 were included in this study (Ethics REC (No.: 041018/8804; Project ID: 0624)). All surgically resected tumor specimens and control specimens were fixed with 10% formalin and embedded in paraffin (FFPE). Samples were staged using The Union for International Cancer Control Tumor-Node-Metastasis (TNM) Classification of Malignant Tumors 8th edition [37,38] and subsequently subtyped histologically using World Health Organization guidelines [39]. Table 2 presents a summary of the available clinical and histopathological data (including age, sex, smoking status, histology, TNM stage, surgical procedure, tumor grade, and primary site) for the patients used in this TMA previously described by us [40,41].

A Beecher Manual Tissue Arrayer (Estigen OÜ, Tartu, Estonia-Model MTA-1) was used to generate a tissue microarray (TMA) containing quadruplicate cores (0.6 mm) of the FFPE embedded samples and a 4 µm section of this TMA was subsequently used for immunohistochemistry (IHC) of JADE2.

### 2.5. Immunohistochemistry

IHC was performed on TMA sections by utilizing a standard protocol to deparaffinize, rehydrate and wash the slides. Subsequently, ULTRA cell conditioning (ULTRA CC1), pH9.1, was used for heat-induced epitope retrieval (HIER). For JADE2 antibody staining, slides were incubated with rabbit polyclonal primary antibody HPA055789 (Atlas Antibodies Bromma, Sweden, Merck—Atlas Antibodies Cat#HPA055789, RRID: AB_2682922) diluted in Roche antibody diluent (Roche Diagnostics 05261899001) (1:10) for 64 min at ambient temperature and stained using the OptiViewTM DAB IHC detection kit (Roche Diagnostics 06396500001) on a Roche/Ventana BenchMark XT processor.

Following IHC, staining was independently assessed by two pathologists blinded to the clinical, pathological and follow-up data. Staining intensity was designated as either 0, 1+, 2+ or 3+, and each tumor section was given an H score between 0–300 = 3(% at 3+) + 2(% at 2+) +1(% at 1+). No samples were observed with an overall H Score of 300.

Tumors with high JADE2 expression were designated as those with an average H score above the median value and low expression below the median. Kaplan–Meier analyses were constructed using Prism 5.01 (GraphPad, San Diego, CA, USA).

### 2.6. RNA Isolation, RT-PCR and qPCR Amplification

Total RNA was isolated and converted to complementary DNA (cDNA) using our previously described methodology [42,43,44]. In brief, total RNA was extracted using TRI reagent (Molecular Research Center, Montgomery Road, OH, USA) according to the manufacturer’s instructions. A sample of 250 ng of this total RNA was then pre-treated to remove contaminating genomic DNA with amplification grade DNase I (Sigma-Aldrich, St. Louis, MO, USA—AMPD1-1KT) according to the manufacturer’s instructions [42,43,44]. ReadyScript^®®^ cDNA Synthesis Mix (Sigma-Aldrich, St. Louis, MO, USA—Cat RDRT) was then used to generate the first strand cDNA according to the manufacturer’s instructions [42,43,44].

Real-Time qPCRs for JADE2 (absolute quantification method) were subsequently conducted on these samples using an Applied Biosystems™ StepOnePlus™ Real-Time PCR System PCR platform (Thermo Fisher Scientific, Waltham, MA, USA—Cat 4376598) and 2x SYBR Green qPCR Master Mix (Bimake, Houston, TX, USA—Cat B21403) using the manufacturer’s protocol in a 2-step qPCR program with a synthesized GBlock for JADE2 as the standard using the following primers:

JADE2 FWD: 5′-ATCTGCGGCAGGACCTAGAG-3′

JADE2 REV: 5′-GAGTTTGCAGATGGCGTGTT-3′

The following cycling parameters were used:

An initial polymerase activation of 95 °C for 2 min followed by 35 cycles of 95 °C 15 s and annealing/amplification at 58 °C for 45 s. A melt curve analysis was conducted at the end of the PCR. The data were analyzed using either the default in-built StepOne software (Version 2.3), exported and graphed using Prism 5.01.

Endpoint RT-PCR to examine all members of the JADE family was conducted using the following primers

JADE1 FWD: 5′-GCAGCCTCTGCAATGAGAAG-3′

JADE1 REV: 5′-GCAGCCTCTGCAATGAGAAG-3′

JADE2 (same primers as above)

JADE3 FWD: 5′-GAGTTTGCAGATGGCGTGTT-3′

JADE3 REV: 5′-GAGTTTGCAGATGGCGTGTT-3′

For Endpoint RT-PCR the following primers were used to amplify 18S rRNA [45] for use as a housekeeping gene/loading control.

18S rRNA Forward 5′-GATGGGCGGCGGAAAATAG-3′

18S rRNA Reverse 5′-GGCGTGGATTCTGCATAATGG-3′

The same amplification protocol was used for RT-PCR as above. All PCR products were gel electrophoresed on either 2% (18S rRNA) or 4% agarose (JADE1-3), and the PCR products were visualized using an LI-COR ODYSSEY FC imaging system (LI-COR, Lincoln, NE, USA).

### 2.7. JADE2 Overexpression Studies

A JADE2 overexpression plasmid was generated by PCR of the full-length JADE2 cDNA with primers incorporating appropriate restriction enzyme sites:

JADE2 3.1(-) EcoRV FWD:

5′-TGCTGGATATCATGGAAGAGAAGAGGCGAAAATACT-3′

JADE2 3.1(-) HindIII REV:

5′-CTTAAGCTTTTAGGAGGCCAGTACGCCCATGCGG-3′.

The PCR products were digested with EcoRV/Hind III, and purified and cloned into similarly digested pCDNA3.1(-) (Thermo Fisher Scientific, Waltham, MA, USA—Cat. No: V795-20).

Confirmation of stable integration into the genome was conducted by endpoint PCR on genomic DNA isolated using a high salt DNA extraction protocol (http://www.protocol-online.org/prot/Protocols/Simplified-DNA-Extraction-from-Cell-or-Tissue-1157.html—accessed on 20 March 2023) using the following primers:

KAN/NEO FWD:

5′-GGCTATGACTGGGCACAACAG-3′

KAN/NEO REV:

5′-CGCTTCAGTGACAACGTCGA-3′

All PCR products were gel electrophoresed on 2% agarose, and the PCR products were visualized using an LI-COR ODYSSEY FC imaging system (LI-COR, Lincoln, NE, USA).

Overexpression studies using this construct were conducted by transfecting NCI-H1975 with either pCDNA3.1(-) Empty Vector Control or with pCDNA3.1(-)—JADE2 using Fugene HD according to the manufacturer’s instructions (Promega, Madison, WI, USA Cat. No: E2311). A mixed pool of stable integrants was obtained by selection with G418 at 1000 µg/mL, and subsequently used for cellular proliferation assays.

### 2.8. Validations of Expression Differences for JADE2 in NSCLC

Validation of altered JADE2 mRNA expression in the TCGA NSCLC cohorts was conducted through Lung Cancer Explorer (LCE) [46] using the comparative analysis setting.

Altered JADE2 protein expression in the TCGA-LUAD and LUSC datasets was interrogated using cProSite [47] to interrogate the datasets of the National Cancer Institute’s Clinical Proteomic Tumor Analysis Consortium (CPTAC). These datasets comprise LUAD—comprising 110 tumors, with 101 tumors paired with normal adjacent tissue samples—and LUSC—comprising 108 tumors with 99 paired normal adjacent tissue—which have global proteome data available for interrogation [48,49].

### 2.9. Survival Analysis for JADE2

To examine whether JADE2 mRNA expression has any prognostic value with respect to patient survival, KM-Plot was utilized [50] using a univariate analysis incorporating a Cox proportional hazards model with median expression as the cut-off.

### 2.10. Associations with Key NSCLC Genes, Oncogenic Driver Mutations and Novel Mutated Genes

Associations between JADE2 and the expression of key genes commonly overexpressed or mutated in NSCLC were examined using TIMER2.0 [51] using the modules Gene_Corr, and Gene_Mutation. The muTarget platform was used to interrogate the TCGA datasets containing RNA-sequencing and mutation data to identify novel mutated genes that result in significant changes in the expression of JADE2 in NSCLC compared to the corresponding wild-type gene expression. The analysis was conducted using JADE2 as the target gene and with mutation prevalence set at 2% [52].

### 2.11. Correlations between JADE2 Expression, Tumor Mutational Burden and Immune Infiltrations in NSCLC

Correlations between tumor mutational burden (TMB) and JADE2 mRNA expression were examined using cBioPortal [53]. To interrogate how altered expression of JADE2 expression was associated with tumor-infiltrating immune cells (TIICs) in NCSLC we used TIMER [54] and TIMER2.0 [51]. TIMER and TIMER2.0 calculate gene expression levels against tumor purity, and results are presented based on a purity-corrected partial Spearman’s rho value with associated statistical significance.

### 2.12. Correlations between JADE2 Expression and Anti-Cancer Drug Sensitivity in NSCLC Cell Lines

The DepMap PRISM repurposing Primary Screen was used to conduct an online analysis restricted to NSCLC cell lines for drugs potentially capable of targeting JADE2 [55].

### 2.13. Drug Treatment and Cellular Viability Assays

Ornidazole was purchased from Selleck (St. Louis, MO, USA; cat. no. O7753), and dissolved in ethanol at a final concentration of 200 mM. Next, 2000 cells were plated per well in a 96-well plate and serum-starved (0.5% *v*/*v* FBS) for 24 h prior to the addition of either drug or vehicle and incubated for a further 24 h. Cellular viability was assessed using a resazurin reduction assay as previously described [45].

### 2.14. Statistical Analysis

All data are expressed as mean ± SEM unless stated otherwise. Statistical analysis was performed with GraphPad Prism 5.01 (GraphPad, La Jolla, CA, USA). Correlations between JADE2 expression and any given parameter were evaluated using the nonparametric Mann–Whitney U-test (for two categories) or Spearman’s correlation as indicated. Kaplan–Meier curves were performed for survival curves, and statistical analysis was assessed using the log-rank (Mantel–Cox) test. Total overall survival (OS) was defined as the time from the date of surgery to death. The 5-year OS was defined as the time from the date of surgery up to a cut-off of five years post-surgery. Patients who were still alive or lost to follow-up were treated as censored data in the survival analysis. Overall, 95% confidence intervals (CIs) were used throughout the analysis. Statistical significance was defined as *p* < 0.05.

Availability of data and materials:

The data that support the findings presented in this study are available for interrogation at the following online resources:TIMER: https://cistrome.shinyapps.io/timer/ (accessed on 19 August 2022)TIMER2.0: http://timer.cistrome.org/ (accessed on 24 August 2022)GEPIA2.0: http://gepia2.cancer-pku.cn/#index (accessed on 29 July 2022)LCE: http://lce.biohpc.swmed.edu/lungcancer/ (accessed on 22 August 2022)KM-PLOT: https://kmplot.com/analysis/index.php?p=background (accessed on 30 August 2022)cBioPortal: https://www.cbioportal.org/ (accessed on 25 August 2022)muTarget: https://www.mutarget.com/ (accessed on 23 August 2022)DepMap: https://depmap.org/portal/ (accessed on 23 August 2022)cProSite: https://cprosite.ccr.cancer.gov/#/ (accessed on 13 July 2022)PROGgeneV2: http://www.progtools.net/gene/ (accessed on 29 August 2022)

## 3. Results

### 3.1. Expression of JADE2 in a Panel of Normal Lung and NSCLC Cell Lines

Utilizing RT-PCR, the expression of JADE1-3 was examined in a panel of normal (HBEC3-5) and NSCLC cell lines comprising cells derived from adenocarcinoma (A549, NCI-H2228, NCI-H1819, NCI-H1975, HCC-827 and NCI-H3122), squamous cell carcinoma (SK-MES-1), large cell carcinoma (NCI-H460) and adenosquamous (NCI-H596) histotypes (Figure 1). All cell lines tested ubiquitously expressed varying levels of JADE mRNAs, with higher basal expression observed predominantly in the NSCLC cancer cells (Figure 1). Given the recent link identified between JADE2 and NSCLC cancer risk [33], we decided to focus on this member of the JADE family in NSCLC.

### 3.2. Expression of JADE2 in Primary NSCLC

We then assessed the expression of JADE2 mRNA in a panel of surgically resected fresh frozen normal/tumor-matched patient samples by qPCR (Figure 2A). Overall, levels of JADE2 mRNA were not significantly altered across this cohort of samples (*p* = 0.2752). We subsequently interrogated a larger cohort of samples namely the TCGA LUAD and LUSC datasets for altered expression of JADE2. When examined, significantly decreased expression of JADE2 mRNA was observed for both LUAD (Figure 2B; *p* = 0.00086) and LUSC (Figure 2C; *p* = 0.022). However, when stratified by stage, no significant differences between stages were observed. Using cProSite [47], we then examined the levels of JADE2 protein in available TCGA samples and the results show that expression of JADE2 protein is not significantly altered in either LUAD (Figure 2D; *p* = 0.4189) or LUSC (Figure 2E; *p* = 0.4794). We also examined the expression of JADE1 and JADE3 at both the mRNA and protein levels in the available TCGA datasets and the results are provided in Appendix A. The mRNA and protein for JADE1 are significantly downregulated in both LUAD and LUSC. JADE3 is significantly upregulated at the mRNA level in LUAD and LUSC, but is only significantly altered at the protein level in LUAD (Appendix A). Finally, using GEPIA we combined the GTEx Normal and TCGA cancer cohorts to conduct a pan-cancer analysis of JADEs1-3 (individually and as a combined signature), and the results are presented in Appendix A). When GTEx normal samples were included, the mRNA for JADE2 was shown to be not significantly altered between tumor and normal in both LUAD and LUSC which correlates with that observed for the cProSite protein data (Appendix A).

### 3.3. Potential Prognostic Value of JADE2 Protein in NSCLC

To assess JADE2 expression for potential clinical value, we optimized immunohistochemistry for JADE2 and developed an H score ranging between 0 and 300, and representative examples of staining/score are shown in Figure 3.

Using KM-Plot we first examined if JADE2 mRNA expression had any prognostic value. High expression of JADE2 mRNA was associated with better overall survival (OS) in NSCLC (Figure 4A; HR = 0.68; *p* = 2 × 10^−9^). When this analysis was stratified according to histological subtype, the OS benefit was restricted solely to LUAD (Figure 4B; HR = 0.53; *p* = 1.3 × 10^−7^) and was not seen in LUSC (Figure 4C; HR = 1; *p* = 0.99). We subsequently examined the expression of JADE2 in patient samples by IHC.

When overall survival was examined there was no apparent survival benefit observed in the patient cohort (Figure 4D; *p* = 0.0971). However, if the analysis was conducted for 5-year OS, high expression of JADE2 was associated with a significant OS benefit (Figure 4E; *p* = 0.0179).

### 3.4. Correlations between JADE2 Expression and Key Genes Associated with NSCLC

Correlations between JADE2 expression and genes commonly overexpressed in NSCLC were examined using TIMER2.0 and the results are presented in Table 3. Using a cut-off of R > 0.33 (positive correlation) or R > −0.33 (negative correlation), significant links between EGFR and PIK3CA with altered JADE2 mRNA expression were observed for both LUAD and LUSC.

We subsequently assessed if the significant dysregulation of JADE2 could be linked to the mutational status of these genes in tumors, and again using TIMER2.0 we examined a number of genes commonly mutated in lung cancer to determine whether mutations within these key genes were correlated with altered JADE2 mRNA expression levels and the results are presented in Table 4. In this regard, no significant changes were observed except for CDKN2A, which if mutated in LUSC resulted in a significant correlation with JADE2 mRNA (Table 4).

### 3.5. Correlations between Novel Mutated Genes and JADE2 mRNA Expression

Using muTarget [52], we then analyzed whether mutations in any other genes may affect JADE2 expression in LUAD and LUSC. From this analysis, 59 genes were identified in LUAD which if mutated resulted in a significant alteration in JADE2 expression, while 21 genes were identified in LUSC and the results of the top five mutated genes as defined by muTarget that affect JADE2 expression for both LUAD (SLC22A25, HECW2, CNTN4, KCNT1, KRT34) and LUSC (TEP1, HECW1, MYOM3, ZNF800, AMER1) are presented in Figure 5, and the full results for mutated genes which affect JADE2 expression in LUSC and LUAD are provided in Appendix A.

### 3.6. Correlation Analysis between JADE2 Expression and Tumor Mutational Burden

Tumor mutational burden (TMB) is widely considered to be a biomarker for predicting potential patient response to immune checkpoint inhibitor (ICI) therapy [56,57]. Using the methodology described by Feng and Shen [57], we, therefore, analyzed the correlation between JADE2 expression and genes associated with either the DNA damage response (DDR) pathway or the mismatch excision repair (MMR) pathways as potential biomarker proxies for TMB and the results are provided in Table 5.

There was no significant correlation between the combined signatures for the DDR pathway in LUAD (*p* = 0.21) whereas this signature showed a strong positive correlation in LUSC (*p* = 4.7 × 10^−6^). The signature for genes associated with the MMR pathway showed significant correlations in LUAD (*p* = 3 × 10^−13^), and also in LUSC (*p* = 1.7 × 10^−4^), suggesting that there may be a correlation between TMB and JADE2 mRNA expression.

As TMB has been associated with response to anti-PD-L1 inhibitors in NSCLC [59], we then examined the correlations between JADE2 mRNA and PD-L1 mRNA expression in LUAD and LUSC. As shown in Figure 6, a significant positive correlation is seen between JADE2 and PD-L1 mRNA expression in LUAD (Figure 6A; *p* = 9.96 × 10^−24^), and also in LUSC (Figure 6B; *p* = 1.54 × 10^−9^), suggesting a potential link between TMB, JADE2 and PD-L1 mRNA expression. However, by querying cBioPortal [53], we observed that TMB is negatively associated with JADE2 mRNA expression in LUAD (Figure 6C; *p* = 1.9 × 10^−7^), whilst there was no association between JADE2 expression and TMB in LUSC (Figure 6D; *p* = 0.382). It must be noted, however, that cBioPortal does not have TMB data on all the patients, and the correlations observed reflect only a subset of patients.

Overall, it would appear that TMB is not associated with JADE2 mRNA expression in NSCLC.

### 3.7. Effects of JADE2 mRNA Expression on Immune Cell Infiltration

To assess the potential impact of decreased JADE2 mRNA expression on tumor immunity, an analysis of tumor-infiltrating immune cells (TIICs) in NSCLC was conducted using TIMER [54]. Following purity adjustment, Spearman’s rho and significance for six immune cell types were generated and the results are presented in Table 6. The results suggest that the decreased expression of JADE2 mRNA in NSCLC was positively correlated with all TIICs examined in LUAD (Table 6a), whereas in contrast, decreased JADE2 mRNA expression in LUSC was significantly positively associated with only CD4+ T cells and dendritic cells (Table 6a).

Subsequently, when JADE2 expression and TIICs were examined for correlations with survival, only B cell and dendritic cell immune infiltrates had survival benefits and this was further restricted to the LUAD subset only (Table 6b).

We re-assessed the effects of JADE2 mRNA on immune infiltrates using TIMER2 [51], which provides a more robust estimation of immune infiltration levels for The Cancer Genome Atlas (TCGA) datasets by using six state-of-the-art algorithms (TIMER, xCell, MCP-counter, CIBERSORT, EPIC and quanTIseq), and the results are presented in Appendix A (LUAD) and Appendix A (LUSC).

### 3.8. Correlation Analysis between JADE2 Expression and Immune Cell Exhaustion

Using TIMER2.0, we assessed the correlations between JADE2 mRNA expression and the expression of important markers of T cell exhaustion [51,60]. The markers chosen were PD-1 (PDCD1), CTLA4, LAG3, TIM-3 (HAVCR2) and GZMB and the results of this analysis are presented in Table 7. After correlation adjustment by purity, JADE2 expression was positively correlated with the expression levels of PD-1 (PDCD1), CTLA4, LAG3 and TIM-3 (HAVCR2) in LUAD (Table 7), while its expression was positively correlated with all five markers in LUSC (Table 7).

A second assessment of T cell exhaustion was subsequently carried out using GEPIA2 [58] which has a pre-defined set of T cell exhaustion markers (PDCD1, HAVCR2, TIGIT, LAG3, CXCL13 and LAYN). A similar pattern to that observed for our analysis in TIMER was observed (Figure 7) where a positive correlation between this 6 gene signature and JADE2 mRNA expression with T cell exhaustion occurs both in LUAD (r = 0.23, *p* = 2.3 × 10^−7^) and in LUSC (r = 0.2, *p* = 6.5 × 10^−6^) as shown in Figure 7.

### 3.9. Identifying Compounds That Can Potentially Target JADE2 in NSCLC

To our knowledge, there are no drugs currently available that specifically target JADE2. To identify any drugs that could potentially be repurposed to target JADE2 in NSCLC we conducted an analysis using the DepMap PRISM repurposing Primary Screen [55] to identify candidate drugs. From this analysis, we identified ornidazole as a primary candidate for investigation (Appendix A). However, when cells were treated with this drug, no significant effects on cellular proliferation were observed (Figure 8A–C), and even when JADE2 was stably overexpressed (Figure 8D,E) in NCI-H1975, no significant alteration to ornidazole sensitivity was observed (Figure 8F).

## 4. Discussion

The role of PHD-domain-containing proteins as potential therapeutic targets in cancer is well established [61]. Many proteins contain PHD domains [9], and one small family of proteins called the JADE family has been shown to play various roles in cancer. During the preparation of this manuscript, a link between a particular SNP (rs329118 T>C) in JADE2 and the risk of developing NSCLC was established [33]. In this manuscript, we examined the expression of JADE2 in NSCLC. Initial analysis of all three JADE mRNAs identified JADE2 as the main JADE family member expressed in NSCLC cell lines (Figure 1).

Yang et al. suggested that JADE2 mRNA was elevated in NSCLC, but our data (Figure 2A) and subsequent analysis of the TCGA datasets (Figure 2B,C) show that overall JADE2 mRNA is either not affected or significantly downregulated in NSCLC tumors. Further analysis showed that JADE2 protein was not significantly altered between adjacent normal lung tissue and LUAD tumors (Figure 2D) or LUSC (Figure 2E). This suggests that JADE2 is not generally overexpressed at either the mRNA or protein level in NSCLC. To assess if JADE2 mRNA had any prognostic value an in silico bioinformatic analysis on JADE2 mRNA expression in NSCLC was conducted using KM-Plotter [50]. From this analysis, it was shown that high expression of JADE2 mRNA was associated with a better OS in all NSCLC (Figure 4A—84 months vs. 52 months). When stratified by histology this OS benefit was restricted to LUAD (Figure 4B), but not LUSC (Figure 4C). When re-analyzed for progression-free survival (PFS) the same results were observed (Appendix A).

While the JADE2 protein does not appear to be significantly altered in NSCLC, its mRNA expression was shown to have potential value to predict both patient OS (Figure 4) and PFS (Appendix A). As such we optimized conditions to conduct IHC and developed a staining intensity scoring (H score) (Figure 3) to allow for an analysis of NSCLC patient TMA for OS. From this initial analysis, it was shown that overall there was no OS benefit (Figure 4D). Traditionally, however, for patients with NSCLC median OS and 5-year survival rates have historically been poor [1]. When re-analyzed for 5-year OS, a significant OS benefit was however observed in patients with high expression of JADE2 (Figure 4E). The disparity between these observations may reflect the age group of the cases examined (Table 2), as the majority of patients were over 65 at the time of surgery and some patients in the full survival dataset may have died as a consequence of other factors (for example old age or other co-morbidities). The data, therefore, suggest that JADE2 protein expression may have potential clinical use for predicting or stratifying patients who will have better 5-year survival.

To determine if JADE2 expression could be associated with other genes known to be important in NSCLC tumorigenesis, in the first instance we assessed if there were any correlations between their overall expression and that of JADE2 mRNA. Positive correlations between JADE2 mRNA expression and some of the examined genes were observed in EGFR and PIK3CA (Table 3) for both LUAD and LUSC. In only one instance was a positive correlation found between ERBB2 mRNA in LUAD, whereas a corresponding correlation was not observed in LUSC (Table 3). This suggests that perhaps JADE2 may play a role in regulating the transcription of these important NSCLC genes but further analysis will be required. As another member of the JADE family JADE1 has been linked to renal cancer pathogenesis through a VHL-mutation-dependent mechanism [62], we next sought to determine if JADE2 mRNA was also associated with mutation of key oncogenic driver mutations in NSCLC such as KRAS or ALK [4]. The results clearly found no associations between the majority of oncogenic drivers, with the exception of CDKN2A which was found to have an association with JADE2 mRNA expression in LUSC (Table 4). This suggests that JADE2 mRNA does not play any significant role in the regulation or stabilization of oncogenic driver mutated mRNA/protein in NSCLC. Using muTarget we subsequently assessed if any genes mutated in NSCLC are affected by JADE2 mRNA expression and identified several such (Figure 5). Several of these such as HECW1 (Figure 5G) or ZNF800 (Figure 5I) have been previously shown to have roles in NSCLC pathogenesis [63,64] or in pre-cancerous settings such as COPD (KRT34, Figure 5E). Whether or not mutated forms of these genes play functional roles in NSCLC tumorigenesis has yet to be elucidated, but preliminary analysis by us suggests that high expression of HECW2 and ZNF800 mRNA is associated with a better OS, whilst high expression of HECW1 and KRT34 is associated with poor OS, and an assessment of these markers for prognostic value linked to mutational status may be warranted.

TMB is a hotly researched area in NSCLC as a high TMB has been linked to better patient responses to ICI [56,57]. To interrogate whether or not JADE2 mRNA is linked to TMB, we first used a surrogate approach first described by Feng and Shen [57], to examine if there were any correlations between JADE2 mRNA and genes associated with both the DDR or MMR repair pathways (Table 5). The results suggest that JADE2 mRNA is positively associated with these pathways and may therefore be associated with TMB. In agreement with this, when analyzed a positive correlation between JADE2 mRNA and PD-L1 mRNA (a key ICI target in NSCLC) was observed for both LUAD (Figure 6A) and LUSC (Figure 6B), suggesting that indeed JADE2 mRNA could correlate with TMB. One of the significant issues in the use of TMB for stratifying patients into those that can receive ICI concerns the methodology to detect TMB (defined as “…the number of somatic mutations per megabase of interrogated genomic sequence” [65]) is the cost per assay [66], and so a simplified assay such as JADE2 mRNA levels that could help predict patient’s sensitivity to ICI would be of potential benefit. However, we then assessed JADE2 mRNA levels with actual TMB in NSCLC, there was an actual negative correlation between JADE2 mRNA and TMB in LUAD (Figure 6C; Spearman’s correlation −0.34; *p* = 1.9 × 10^−7^), and no correlation between TMB and JADE2 mRNA in LUSC (Figure 6D).

Another potential indicator for response to ICI in NSCLC is TIICs and, in particular, CD8+ T cell infiltration [67]. Analysis of the TCGA datasets for TIICs that correlate with JADE2 mRNA expression identified that at least in LUAD JADE2 mRNA was strongly associated with CD8+ T cell infiltration in NSCLC (Table 6a). Significant associations for many other TIICs were also observed for LUAD, in particular, CD4+ T cells, neutrophils and dendritic cells (Table 6a), while few associations were observed for LUSC with the exception of CD4+ T cells and dendritic cells (Table 6a). When associated with patient survival, only B cell and dendritic cell infiltration were found to have any survival benefit in LUAD (Table 6b). Immune cell exhaustion is also associated with resistance or lack of response to ICI [67,68]. We subsequently assessed whether JADE2 mRNA was associated with well-established markers of T Cell exhaustion and observed positive correlations between JADE2 mRNA and markers of immune cell exhaustion (Table 7), and a second analysis using a different methodological approach subsequently confirmed that a six-gene signature for markers of T cell exhaustion is positively correlated with JADE2 mRNA (Figure 7). Overall, the results described above indicate that JADE2 mRNA may not be suitable as a candidate biomarker to predict overall response to ICI in NSCLC; however, future studies will be required to delineate its roles in CD8+ T cell infiltration and regulation of PD-L1 expression and to assess it as a potential biomarker to predict T cell exhaustion in NSCLC.

Ornidazole an antiprotozoal antibiotic was identified through analysis of the Cancer Dependency Map [55] as a potential drug that could be repurposed for use in NSCLC. In this regard, ornidazole was previously studied as a modifier of hypoxia in various cancer clinical trials [69,70], with no significant improvement observed for lung cancer [70]. However, the peak plasma concentration of ornidazole observed in the astrocyte study was 40 mg/l [70], which equates to approximately 180 µM. Interestingly the DepMap data which suggested that JADE2 mRNA was associated with sensitivity to ornidazole used an 8-step, 4-fold dilution of ornidazole, starting from 10 µM. Our results for sensitivity to ornidazole demonstrated that at least in concentrations similar to those used in earlier trials that there was some sensitivity to ornidazole at the 150–200 µM range (and beyond). However, this sensitivity was also observable in normal bronchial epithelial cells, suggesting that as a stand-alone repurposing agent, ornidazole should not be considered suitable for further evaluation in NSCLC. To further test if JADE2 expression had any role in sensitivity to ornidazole a cell line stably overexpressing JADE2 was generated (Figure 8D,E), and when sensitivity to this drug was assessed overexpression of JADE2 did not affect sensitivity (Figure 8F). It must be noted that the methodology used to measure the effects on cellular proliferation by us and that used by DepMap for their chemical-perturbation viability screens is very different, and may require additional refinements/re-assessments to determine if ornidazole could be repurposed (maybe in combination with standard chemotherapy) in NSCLC. Moreover, a recent study in melanoma suggests that ornidazole could play a role in suppressing CD133+ stem cells [71], and CD133+ is commonly expressed in NSCLC [72,73], and be a potential novel target [74,75].

## 5. Conclusions

In conclusion, these results suggest that whilst JADE2 mRNA and protein are not significantly altered in NSCLC, their expression has prognostic value, and may have potential clinical use for predicting or stratifying patients who will have better 5-year overall survival.

Several areas of investigation will be required moving forwards to critically delineate the potential roles of JADE2 in cancer. Such areas should include:Knockdown or CRIPSR-based editing of JADE2 expression. In addition, it is potentially conceivable that redundancy between other JADE family members could substitute for JADE2, so studies on JADE1 or JADE3 will also be necessary.In vivo animal studies will be required to assess if overexpression of JADE2 affects NSCLC tumourigenicity/aggressiveness.Given the limited effects of ornidazole when tested as a “stand-alone” agent, combinatorial treatments of standard NSCLC therapies with ornidazole are warranted to assess if this agent can either enhance therapy or resensitize resistant cells to therapy, or affect the expression of checkpoint inhibitor targets.Studies to assess whether ornidazole can affect cancer stem cell populations in NSCLC are warranted given the data emerging from melanoma [71]. In this regard, it may be possible to test this in a panel of isogenic parent/cisplatin-resistant cell lines which we have previously generated and shown to be enriched for CD133+ cells [76].

## Figures and Tables

**Figure 1 biomedicines-11-02576-f001:**
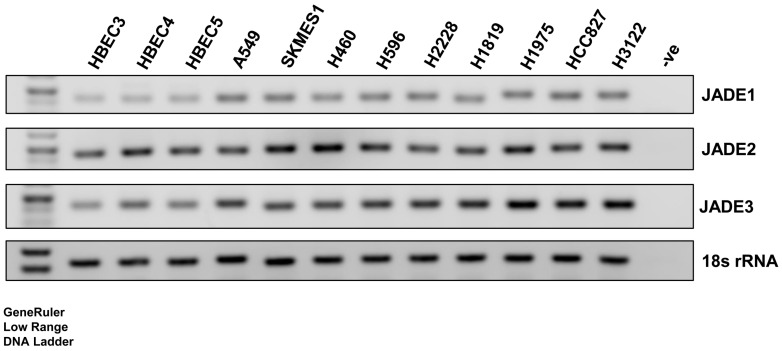
Expression of JADE1-3 in a panel of NSCLC cell lines. Endpoint RT-PCR was used to assess the expression of JADE mRNAs in a panel of lung cell lines including normal human bronchial epithelial cells (HBEC3, HBEC4 and HBEC5) alongside NSCLC cell lines. 18S rRNA was included as a loading control.

**Figure 2 biomedicines-11-02576-f002:**
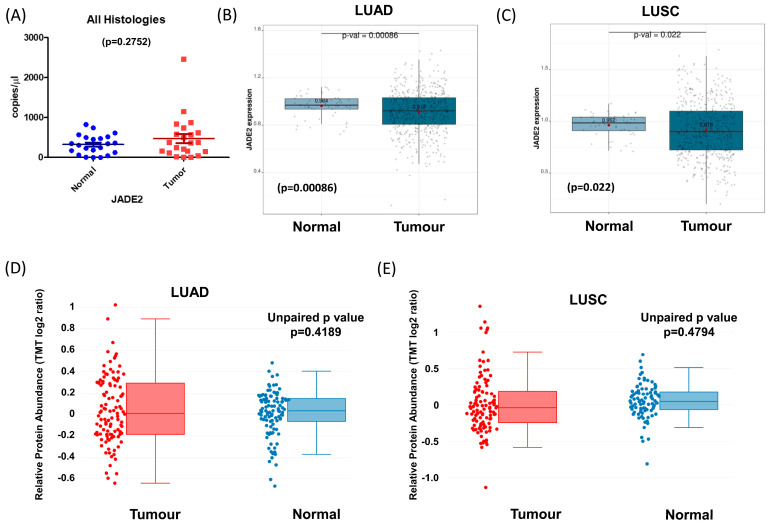
Altered expression of JADE2 in NSCLC. Analysis of expression of JADE2 in NSCLC. Examination of changes to JADE2 mRNA levels in fresh frozen surgically resected patient samples comprising (**A**) all histologies; (**B**) comparative analysis of *JADE2* mRNA levels in The Cancer Genome Atlas (TCGA) Lung Adenocarcinoma (LUAD) and (**C**) the Lung Squamous Cell Carcinoma (LUSC) datasets using Lung Cancer Explorer (LCE) [46]. (**D**) Expression of JADE2 total protein levels in LUAD and (**E**) expression of JADE2 total protein levels in LUSC as assessed using cProSite [47,48,49]. * *p* < 0.05; *** *p* < 0.001.

**Figure 3 biomedicines-11-02576-f003:**
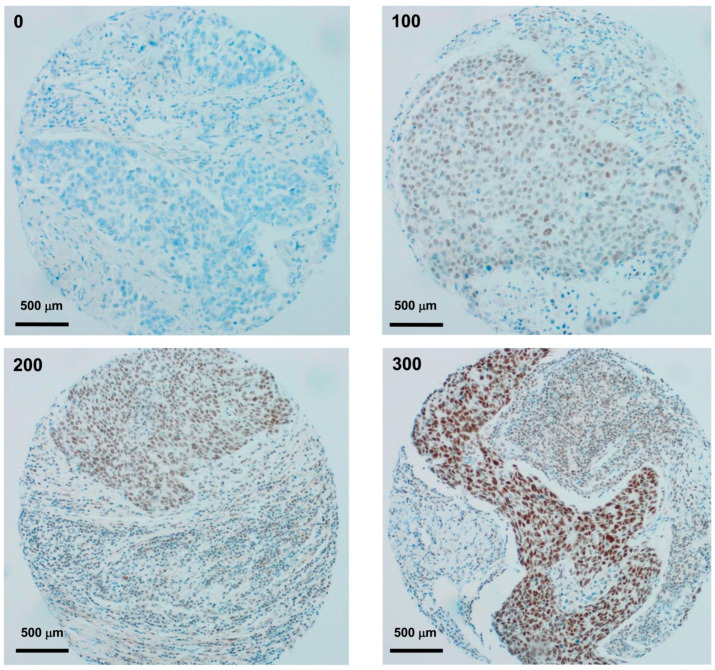
JADE2 IHC scoring intensity. Representative examples of JADE2 IHC stained clinical NSCLC samples, displaying varying levels of JADE2 expression (40× magnification) as per the assigned H Score running from negative staining (H score 0); low level of expression (H score 100); medium level of expression (H score 200) to high level of expression (H score 300) in patient TMA cores. However, it must be noted that as the H scores were averaged over available cores per patient an overall final H score of 300 was never achieved.

**Figure 4 biomedicines-11-02576-f004:**
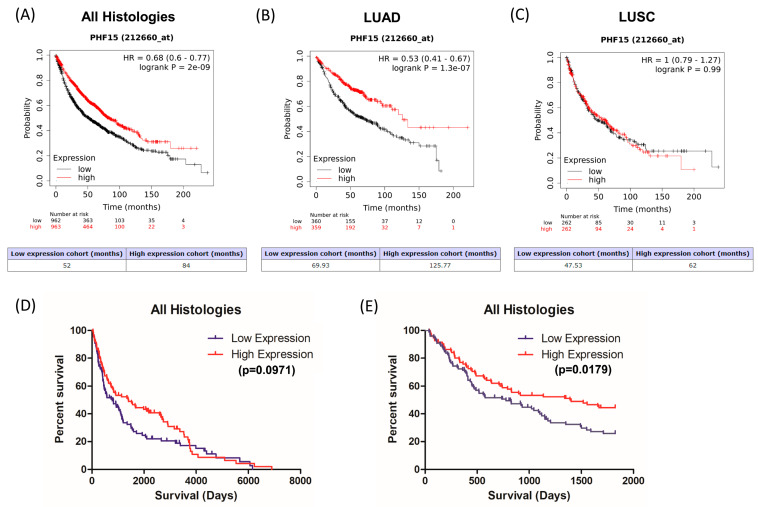
Prognostic value of JADE2 in NSCLC. The prognostic value of *JADE2* mRNA expression was assessed for overall survival (OS) using KM-Plotter [50] and by IHC on a patient TMA. Higher expression of the mRNA for *JADE2* was associated with better OS overall (**A**); which, when stratified by tumor histology, was limited to the LUAD subtype only (**B**); whilst no difference in OS was observed for LUSC (**C**). When examined by IHC there was no significant OS benefit observed for high JADE2 protein expression for overall patient survival (**D**), but when 5-year OS was calculated a significant OS benefit was observed for patients with high JADE2 protein compared to those with low expression (**E**). *p* < 0.05 was considered to be significant.

**Figure 5 biomedicines-11-02576-f005:**
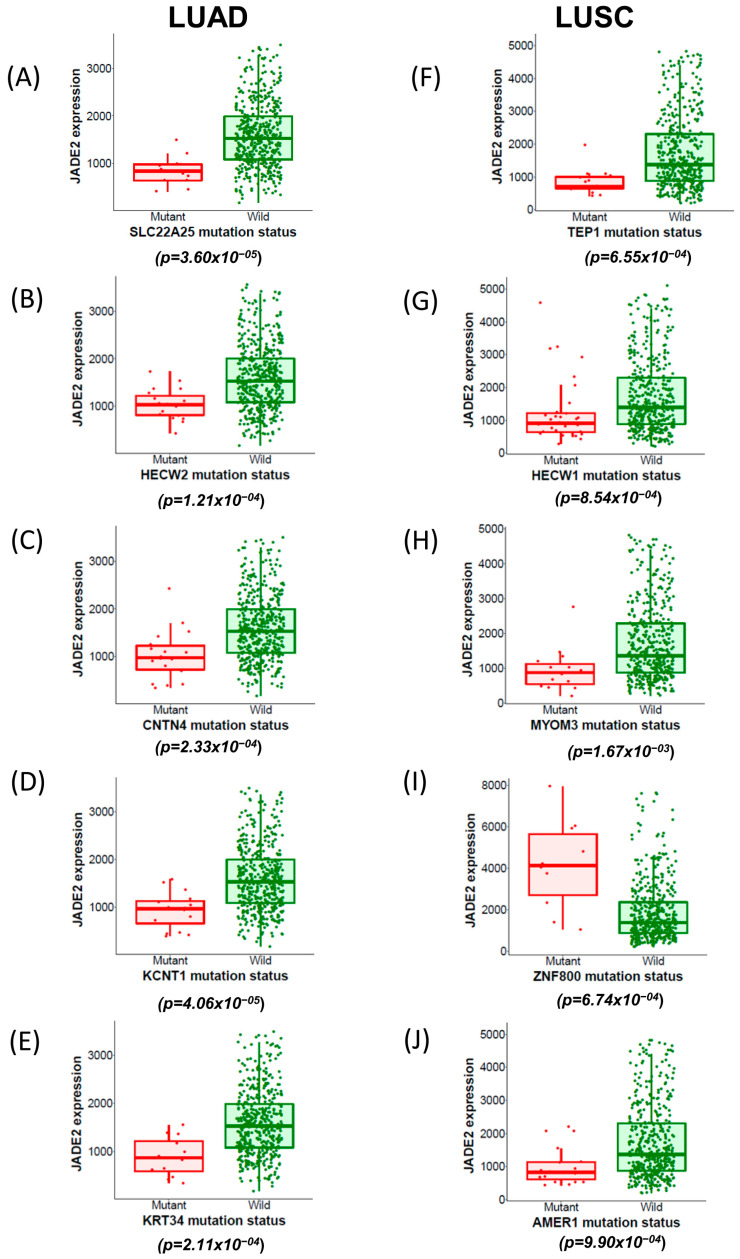
Identification of mutated genes that affect JADE2 expression in NSCLC. JADE2 mRNA expression changes and mutation status in NSCLC were examined using muTarget [52]. The resulting analysis identified several genes which, if mutated, resulted in significantly altered JADE2 mRNA expression as follows: (**A**) SLC22A25, (**B**) HECW2, (**C**) CNTN4, (**D**) KCNT1 and (**E**) KRT34 in LUAD; and (**F**) TEP1, (**G**) HECW1, (**H**) MYOM, (**I**) ZNF800 and (**J**) AMER1 in LUSC.

**Figure 6 biomedicines-11-02576-f006:**
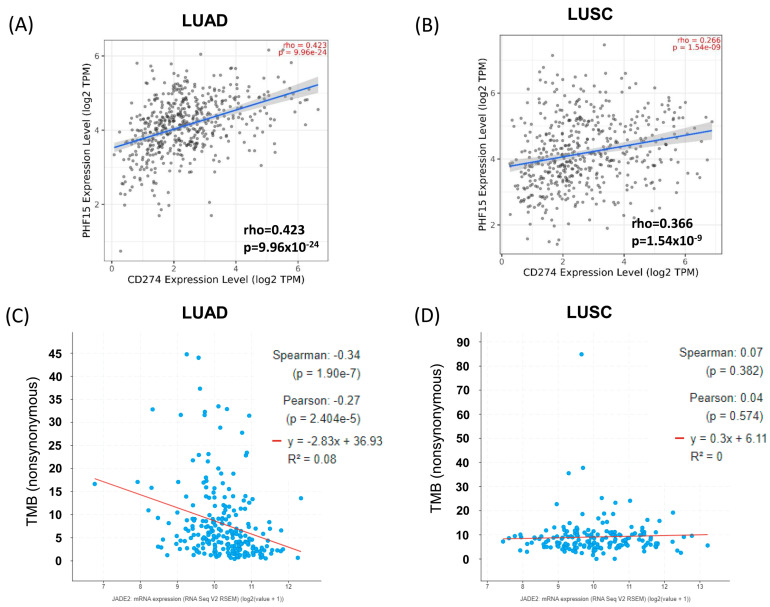
Correlations of JADE2 mRNA expression and tumor mutational burden. Correlations between JADE2 mRNA expression and PD-L1 mRNA expression were examined in (**A**) LUAD and (**B**) LUSC using TIMER2.0 [51]. Subsequently, cBioPortal was used to examine the correlations between TMB and JADE2 mRNA in (**C**) LUAD and (**D**) LUSC.

**Figure 7 biomedicines-11-02576-f007:**
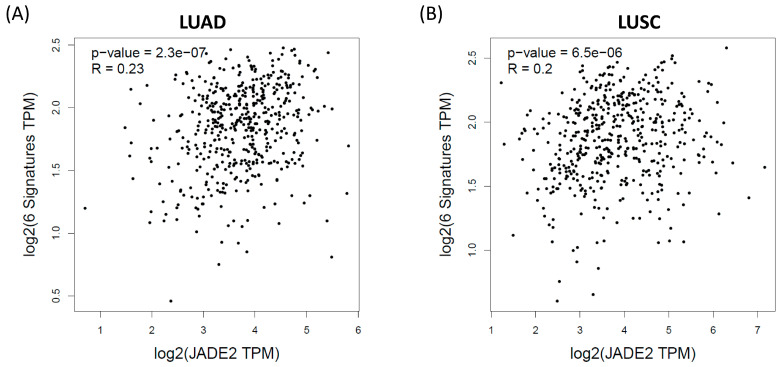
Correlations between JADE2 mRNA and signatures of immune cell exhaustion. GEPIA2 [58] was used to assess whether there were any correlations between JADE2 mRNA expression and a six-gene signature of immune cell exhaustion in (**A**) LUAD and (**B**) LUSC.

**Figure 8 biomedicines-11-02576-f008:**
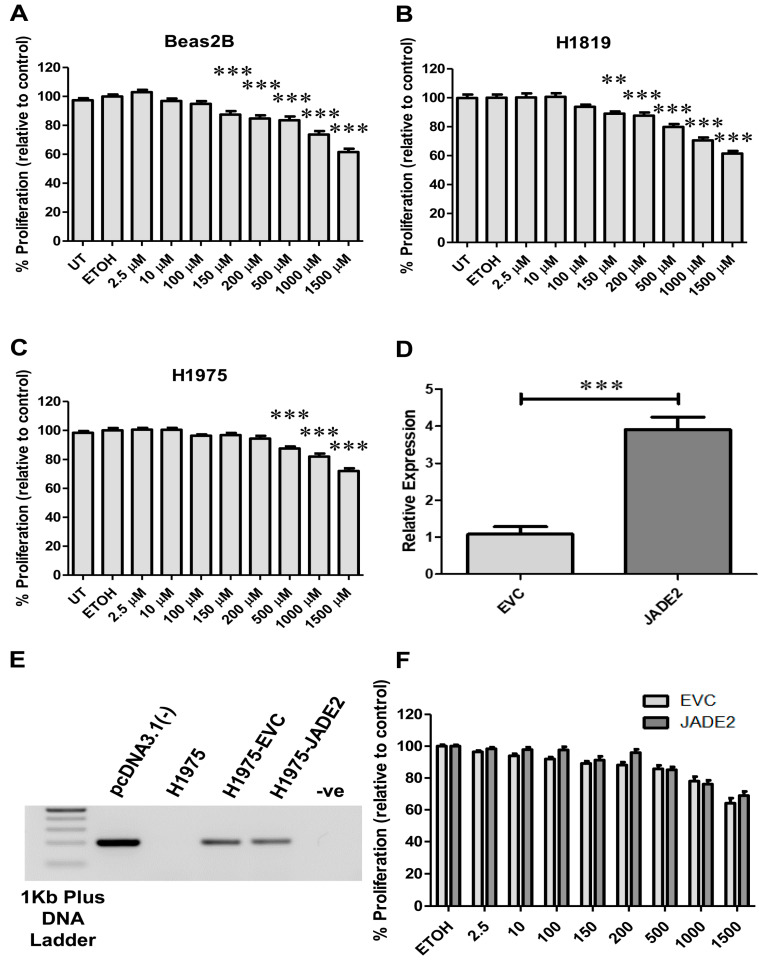
Identification and testing of ornidazole, a candidate drug that can target JADE2 in NSCLC cell lines for anti-cancer drug sensitivity. Cells were treated for 24 h with increasing concentrations of ornidazole and effects on cellular proliferation were assessed using resazurin as follows: normal bronchial cells (BEAS2B) (**A**); NCI_H1819 (**B**); and NCI-H1975 (**C**). Stable overexpression cells were generated in NCI-H1975 and tested for increased mRNA expression (**D**) and stable integration into the genome (**E**). The effects of whether overexpression altered sensitivity to ornidazole were subsequently assessed on cellular proliferation (**F**). Statistical significance was set at ** *p* < 0.01; *** *p* < 0.001.

**Table 1 biomedicines-11-02576-t001:** Details of surgically resected fresh frozen patient samples used in this study.

Sample	Histology	Sex	Age	Stage (7th Edition)	TNM
1	Adenocarcinoma	Female	75	IV	pT4 N2 M1a
2	Adenocarcinoma	Male	71	IA	pT1a N0
3	Adenocarcinoma	Female	75	IIA	pT1a N1
4	Adenocarcinoma	Male	71	IB	pT2a
5	Adenocarcinoma	Female	78	IB	pT2a
6	Adenocarcinoma	Female	67	IIIB	pT4 N2
7	Adenocarcinoma	Female	66	IB	pT2a N0
8	Adenocarcinoma	Female	69	IB	pT2a N0
9	Adenocarcinoma	Male	66	IIIA	pT2a N0
10	Adenocarcinoma	Male	86	IIIA	pT3 N1
11	Adenocarcinoma	Male	69	IIIA	pT3 N1
12	Squamous Cell Carcinoma	Female	67	IB	pT2a N0 IB
13	Squamous Cell Carcinoma	Male	71	IB	pT2a N0
14	Squamous Cell Carcinoma	Female	59	IIA	pT2a N1
15	Squamous Cell Carcinoma	Female	66	IIA	pT2a N1
16	Squamous Cell Carcinoma	Male	78	IIA	pT1b N1
17	Squamous Cell Carcinoma	Male	79	IIIA	pT3 N2
18	Squamous Cell Carcinoma	Male	70	IB	T2 N0
19	Squamous Cell Carcinoma	Female	80	IIA	pT2a N1
20	Squamous Cell Carcinoma	Male	72	IIB	pT2b N1
21	Squamous Cell Carcinoma	Male	66	IIIA	pT1b N2
22	Squamous Cell Carcinoma	Female	76	IA	pT1b N0

**Table 2 biomedicines-11-02576-t002:** Patient characteristics in the SJH NSCLC TMA.

	*n*
LUSC	108
LUAD	82
Pleomorphic carcinoma	7
Large cell	3
Adenosquamous	4
Female	79
Male	125
Age < 65	92
Age ≥ 65	112
Node positive	89
Node negative	115
Tumor size ≥ 5 cm	82
Tumor size < 5 cm	122
Grade 1	16
Grade 2	110
Grade 3	78
Stage I	100
Stage II	49
Stage III	54
Stage IV	1
Smoker	100
Ex-smoker	78
Never smoker	26

**Table 3 biomedicines-11-02576-t003:** Correlations between *JADE2* mRNA expression and key genes in NSCLC.

		LUAD		LUSC	
Parameter	Gene	Partial cor.	Adj. *p*-Value	Partial cor.	Adj. *p*-Value
JADE2 expression correlated with	TP53	0.082710513	0.178404525	0.081198214	0.179881123
	KRAS	0.27347762	2.21 × 10^−9^	0.194184439	4.33 × 10^−5^
	EGFR	0.472416177	8.97 × 10^−28^	0.308070729	2.01 × 10^−11^
	ERBB2	0.265614636	1.19 × 10^−8^	0.030231142	0.728719461
	PI3KCA	0.397114574	1.51 × 10^−19^	0.326686247	5.30 × 10^−13^
	ALK	0.234086121	1.46 × 10^−6^	0.119036922	0.019498159

Analysis was conducted using TIMER 2.0 [51]. Results are presented as purity-corrected partial correlation Spearman’s rho value and statistical significance. Correlation cut-off values of the Spearman coefficient were set to R > 0.33 (positive correlation) and R > −0.33 (negative correlation). Analysis conducted on 22 August 2022.

**Table 4 biomedicines-11-02576-t004:** Correlations between *JADE2* mRNA expression and mutation of key genes in NSCLC. Gene Expression Correlations.

		LUAD		LUSC	
	Mutated Gene	log2 Fold Changes	Adj. *p*-Value	log2 Fold Changes	Adj. *p*-Value
JADE2	TP53	−0.051608347	0.163694231	0.064624702	0.163694231
	KRAS	0.043519348	0.408863297	−0.000381679	0.891865206
	EGFR	0.055108336	0.541222696	0.043119948	0.937619343
	ERBB2	−0.080435586	0.796276167	−0.200601312	0.609981468
	PIK3CA	0.061946812	0.648789762	0.006501538	0.945101486
	ALK	0.055475718	0.728248885	-0.04583388	0.884587181
	ROS1	−0.068125014	0.561008551	−0.093883364	0.468144673
	CDKN2A	0.052081676	0.861629309	0.182166956	0.002894015
	PTEN	−0.081612099	0.71248883	0.050782786	0.71248883
	BRAF	−0.05377631	0.610383827	−0.073436266	0.835224651
	MET	0.052563434	0.868825624	−0.132008367	0.868825624
	NF1	−0.043139679	0.792161892	−0.028611057	0.833802286

Analysis was conducted using TIMER 2.0. log2 fold changes for the differential expression of each gene are provided along with the adjusted *p* values [51]. Analysis conducted on 29 August 2022.

**Table 5 biomedicines-11-02576-t005:** Correlation between JADE2 expression and proxy markers of tumor mutational burden.

		LUAD		LUSC	
	Variable	R	*p*-Value	R	*p*-Value
DNA damage response (DDR) pathway	BRCA1	0.1	0.026 *	0.34	1.2 × 10^−4^ ***
	ATM	0.42	1.3 × 10^−22^ ***	0.18	5.2 × 10^−5^ ***
	ATR	0.15	8 × 10^−4^ ***	0.18	5 × 10^−5^ ***
	CDK1	−0.17	1.8 × 10^−3^ ***	0.085	0.065
	CHEK1	−0.078	0.087	0.083	0.069
	CHEK2	−0.22	7.7 × 10^−7^ ***	0.012	0.79
	TP53	0.17	1.3 × 10^−4^ ***	0.1	0.028 *
	Combined Signature	0.057	0.21	0.21	4.7 × 10^−6^ ***
Mismatch excision repair (MMR) related genes	PMS2	0.32	1 × 10^−12^ ***	0.19	3.6 × 10^−5^ ***
	MLH1	0.39	2 × 10^−19^ ***	0.14	2.1 × 10^−3^ ***
	MSH2	0.16	3.3 × 10^−4^ ***	0.064	0.16
	MSH3	0.52	7.8 × 10^−35^ ***	0.21	2.2 × 10^−6^ ***
	MSH6	0.27	8.1 × 10^−10^ ***	0.23	2.3 × 10^−7^ ***
	PCNA	−0.19	0.013 *	0.053	0.24
	Combined Signature	0.32	3 × 10^−13^ ***	0.17	1.7 × 10^−4^ ***

Analysis was conducted using GEPIA2 [58]. Results are presented as Spearman’s rho value (R) alongside statistical significance. * *p* < 0.05; *** *p* < 0.001. Analysis conducted 24 August 2022.

**Table 6 biomedicines-11-02576-t006:** Correlations between JADE2 and immune infiltrations in NSCLC.

(a) Gene Correlations					
		LUAD		LUSC	
	Variable	Partial cor.	*p*-Value	Partial cor.	*p*-Value
JADE2	Purity	−0.180664434	5.38 × 10^−5^ ***	−0.102110571	0.025584713 *
	B cell	0.267772037	2.16 × 10^−9^ ***	−0.008044225	0.86162598
	CD8+ T cell	0.221168991	8.25 × 10^−7^ ***	0.076819743	0.094463581
	CD4+ T cell	0.397473929	9.08 × 10^−20^ ***	0.171171466	0.000177919 ***
	Macrophage	0.21693886	1.42 × 10^−6^ ***	−0.014194577	0.757156887
	Neutrophil	0.301271192	1.36 × 10^−11^ ***	0.086648578	0.058887476
	Dendritic cell	0.397200525	6.83 × 10^−20^ ***	0.166597378	0.000269484 ***
(b) Survival					
		LUAD		LUSC	
	Variable	*p*-Value		*p*-Value	
JADE2	B cell	0.000268218 ***		0.778203963	
	CD8+ T cell	0.345905392		0.370701923	
	CD4+ T cell	0.507773		0.142871314	
	Macrophage	0.110109126		0.651047592	
	Neutrophil	0.081068767		0.126999461	
	Dendritic cell	0.047523634 *		0.324066598	
	JADE2	0.901934355		0.298085102	

Analysis was conducted using TIMER [54]. Results are presented as purity-corrected partial Spearman’s rho value and statistical significance. * *p* < 0.05; *** *p* < 0.001. Partial Cor. partial correlation (partial Spearman’s rho value). Analysis conducted 24 August 2022.

**Table 7 biomedicines-11-02576-t007:** Correlation analysis between JADE2 and markers of immune cell exhaustion in TIMER 2.0.

	LUAD			LUSC		
	Partial cor.	*p*-Value	Adj. *p*-Value	Partial cor.	*p*-Value	Adj. *p*-Value
PD-1 (PDCD1)	0.222296	6.17 × 10^−7^	2.06 × 10^−6^	0.259232	9.17 × 10^−9^	5.24 × 10^−8^
CTLA4	0.200917	6.94 × 10^−6^	2.31 × 10^−5^	0.238258	1.39 × 10^−7^	6.97 × 10^−7^
LAG3	0.122737	0.00636	0.014134	0.228991	4.29 × 10^−7^	2.45 × 10^−6^
TIM-3 (HAVCR2)	0.250338	1.76 × 10^−8^	5.86 × 10^−8^	0.228991	4.29 × 10^−7^	2.45 × 10^−6^
GZMB	−0.03006	0.50554	0.623764	0.153332	0.00078	0.003119

Analysis was conducted using TIMER 2.0 [51]. Results are presented as purity-corrected partial Spearman’s rho value and statistical significance. Neg—negative correlation (*p* < 0.05, *p* < 0); Pos—Positive correlation (*p* < 0.05, *p* > 0); ns—not significant (*p* > 0.05). Analysis conducted 29 August 2022.

## Data Availability

Publicly available datasets were analyzed in this study. These data can be found here: TIMER: https://cistrome.shinyapps.io/timer/ (accessed on 25 June 2023); TIMER2.0: http://timer.cistrome.org/ (accessed on 25 June 2023); GEPIA2.0: http://gepia2.cancer-pku.cn/#index (accessed on 25 June 2023); LCE: http://lce.biohpc.swmed.edu/lungcancer/ (accessed on 25 June 2023); KM-PLOT: https://kmplot.com/analysis/index.php?p=background (accessed on 25 June 2023); cBioPortal: https://www.cbioportal.org/ (accessed on 25 June 2023); muTarget: https://www.mutarget.com/ (accessed on 25 June 2023); DepMap: https://depmap.org/portal/ (accessed on 25 June 2023); cProSite: https://cprosite.ccr.cancer.gov/#/ (accessed on 25 June 2023); PROGgeneV2: http://www.progtools.net/gene/ (accessed on 25 June 2023). Additional data presented in this study are available on request from the corresponding author. The data are not publicly available due to privacy/GDPR restrictions.

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
