# Peer review of "An Analysis of JADE2 in Non-Small Cell Lung Cancer (NSCLC)"

_biomedicines, 2023, doi:10.3390/biomedicines11092576_

Round 1

Reviewer 1 Report

The findings appear to be interesting. Major points that the authors need to address are as follows:

1. What causes altered expression of JADE2 and how it can regulate various hallmarks of cancer in NSCLC is not clear?

2. The rationale for focussing only on NSCLC is not clear.

3. A limited in vivo study will greatly increase the impact of the findings.

4. The mechanisms by which overexpression of JADE2 can modulate a number of genes linked to tumor invasion and metastasis. It should be analysed if overexpression of JADE2 can affect epithelial mesenchymal transition process.

5. The manuscript should be carefully checked for typographical errors.

Needs improvement.

Author Response

Reviewer 1

  1. What causes altered expression of JADE2 and how it can regulate various hallmarks of cancer in NSCLC is not clear?

We thank the reviewer for raising this point. We have now attempted to address this point in the Introduction. The available data for JADE2 (or for JADE1 and JADE3 for that matter) are pretty much non-existent in NSCLC. This is the central tenet of the manuscript, which at its heart is to examine JADE2 in NSCLC. We chose this member of the family on the basis that a SNP in the gene for JADE2 has been associated with an increased risk of NSCLC (unfortunately that sentence was not completed properly in our original draft for which we apologise. In this new revision, we have now provided an assessment of JADE1-3 in a NSCLC cell line panel, and added text to describe the potential link between JADE2 and risk of NSCLC as the rationale for examining JADE2 specifically. We hope that the amended text and Figure will prove sufficient for this reviewer.

  1. The rationale for focussing only on NSCLC is not clear.

We thank the reviewer for raising this. There was an error in the text where a link between JADE2 and risk of developing lung cancer was missing some text on line 96. This has now been corrected to “Most recently, a link between a single nucleotide polymorphism (SNP) in JADE2 and NSCLC cancer risk has been identified”. We have also tried to add in text to the introduction to clearly state that there is a paucity of knowledge regarding the expression and role of the JADE family in lung cancer, and hope that this revised text will be sufficient for Reviewer 1.

  1. A limited in vivo study will greatly increase the impact of the findings.

We thank the reviewer for raising this and agree completely with that sentiment. Unfortunately, lack of funding prevents us from conducting such a study. Should we be able to secure funding in the future we would hope to conduct a xenograft study using the stable overexpression cells compared to stably integrated empty vector cells, to determine if JADE2 overexpression does play role in tumorigenesis. However, at this point in time we are unable to conduct such an experiment and hope that the reviewer understands our point of view in this regard.

  1. The mechanisms by which overexpression of JADE2 can modulate a number of genes linked to tumor invasion and metastasis. It should be analysed if overexpression of JADE2 can affect epithelial mesenchymal transition process.

We appreciate the reviewer’s comments. The mention of tumor invasion and metastasis was an error based on a previous draft and should not have been included in this version. We have some preliminary data from a PCR based array from “Real-Time Primers”. Unfortunately, at this point in time we do not trust this array to give reliable results, and as such had intended to drop this subsection from the manuscript. This is something which we may return to in the future, should funding be obtained to obtain a more robust array (perhaps those from Qiagen), and also to do additional experiments. We believe that despite this data, the existing data presented remains publishable, and hope that the reviewer can accept this explanation.

  1. The manuscript should be carefully checked for typographical errors.

We have now carefully checked the manuscript again for any typographical errors, and hope that any errors have now been located and corrected.

Reviewer 2 Report

This manuscript deals with the analysis of expression of JADE2 protein in non-small cell lung cancer (NSCLC). They stated that high expression of JADE2 is associated with a better 5-year survival. Also, overexpression of JADE2 is associated with genes involved in the tumor invasion and metastasis. They defined some drugs targeting JADE2 as well.

The experiments are performed on a series of public data to characterize the relevance of JADE2, and they performed some in vitro experiments.

Lines 312-314. It is not clear the tissue from which the cell lines have been derived. Indeed, in the title, it is indicated that “malignant NSCLC” will be considered. In the text, it is indicated that mesothelioma cell lines have been considered.  Please specify for each cell line analyzed the correct histotype.  Furthermore, I would say that malignant should be erased. Indeed, NSCLC and mesotehlioma are tumors and so the word malignant is not necessary.

How do the Authors justify the differences found for the level of expression of JADE2 in the surgical cohort compared to those present as databases?

It is evident from the figure 5 that in all the instances, except for the ZNF800, the expression of JADE2 is lower in the mutated group compared to the WT group. Is there any explanation for this difference?

The identification of ornidazole as a potential drug that can inhibit JADE2 activity is not conclusive. Actually, this drug is not working specifically of tumor cells or better in cells overexpressing JADE2. The attempt of the authors to assess the effect of the ornidazole is valuable. The expression of JADE2 on the cell line H1819 is not reported (at least in the figure 1). Why the authors tested this cell line? It should be better to test cells expressing or not the JADE2 for the sensitivity to the drug. Furthermore, it is not so complicated to test the ornidazole at very low doses with other drugs for treatment of lung cancer to identify, perhaps, a specific effect.

Silencing experiments can better define the role of JADE2. Also, the analysis of expression of the other JADE component of the family can shed some new insight. This reviewer does not know if the other members can substitute JADE2 in some of its functions. 

Overall, the manuscript is well written and the effort of the authors is evident. However, the message is weak and the relevance of JADE2 in regulating cell proliferation or other tumor cell features is not clearly demonstrated. In other words, the data shown are not conclusive at all.

Author Response

Reviewer 2

This manuscript deals with the analysis of expression of JADE2 protein in non-small cell lung cancer (NSCLC). They stated that high expression of JADE2 is associated with a better 5-year survival. Also, overexpression of JADE2 is associated with genes involved in the tumor invasion and metastasis. They defined some drugs targeting JADE2 as well.

We apologise for the text regarding tumor invasion and metastasis. This was left in from an earlier version of the manuscript. At the present moment whilst we do believe that JADE2 is involved in regulating the expression of such genes, we have concerns regarding the reliability of the array used to assess these, and in this version of the manuscript have removed any mention of “tumor invasion and metastasis”. This is something that we hope to resolve in the future, and hope that the reviewr can accept these revisions in the present manuscript.

The experiments are performed on a series of public data to characterize the relevance of JADE2, and they performed some in vitro experiments.

Lines 312-314. It is not clear the tissue from which the cell lines have been derived. Indeed, in the title, it is indicated that “malignant NSCLC” will be considered. In the text, it is indicated that mesothelioma cell lines have been considered.  Please specify for each cell line analyzed the correct histotype.  Furthermore, I would say that malignant should be erased. Indeed, NSCLC and mesotehlioma are tumors and so the word malignant is not necessary.

We thank the reviewer for raising this point and apologise for any confusion arising. To address this, we have now added text to both the materials and methods and to lines 312-314 expanding on and clarifying the histotypes of the various cell lines used in the study and figures. Moreover, the text regarding mesothelioma has now been removed from the manuscript as we feel that it is causing confusion, and diluting the overall story. In the future we intend to expand on the JADE family expression and role in mesothelioma, but this will be a separate publication. We hope that the changes made will now make the text clearer to the reviewer and hope that this will prove satisfactory.

How do the Authors justify the differences found for the level of expression of JADE2 in the surgical cohort compared to those present as databases?

Put simply, it reflects a matter of scale. Our cohort of surgically resected patients is limited to 22 matched pairs, the TCGA LUAD dataset (for mRNA) comprises n=59 normal and n=517 tumors while the TCGA LUSC dataset (for mRNA) comprises n=51 normal and n=501 tumors. The cProSite LUAD proteomic dataset comprise n= 110 tumors and 101 matched normal adjacent tissue, while the cProSite LUSC dataset comprises 108 tumors and 99 paired normal adjacent tissue. As such we feel that while differences between mRNA and protein emerge this reflects the differences in the numbers of samples examined, but do not feel that it affects the overall story being presented. Moreover, when it comes to OS, the mRNA data from KM-Plot reflects the data we observe at the protein level by IHC (Figure 4A versus Figure 4E). Moreover, we have now added in a separate pan-cancer analysis which includes the Genotype - Tissue Expression (GTEx) normal tissue expression data. These results are provided as Supplementary Figures, but serve to resolve the issue in that when combined overall the expression of JADE2 in LUAD and LUSC is not significantly altered between tumor and normal, which is reflected in the proteomic data. As such we hope that this explanation goes some way to alleviating any concerns of the reviewer with respect to differences observed in JADE2 expression in NSCLC.

It is evident from the figure 5 that in all the instances, except for the ZNF800, the expression of JADE2 is lower in the mutated group compared to the WT group. Is there any explanation for this difference?

The explanation is simply that the output from MuTarget presents only the top 5 most altered (whether up or down). In this instance the vast majority were associated with decreased expression of JADE2 in the mutated group. We have now included the entire results of the MuTarget analysis as separate tables in the Supplementary material, and hope that this will be sufficient.

The identification of ornidazole as a potential drug that can inhibit JADE2 activity is not conclusive. Actually, this drug is not working specifically of tumor cells or better in cells overexpressing JADE2.

We thank the reviewers for raising this excellent point, and are completely in agreement with the reviewer. It is our opinion that this drug is not working as identified by the DepMap project which used an 8-step, 4-fold dilution of ornidazole, starting from 10 µM. We chose to use a dilution range which included this but also up to the original lung cancer clinical trials which equates to approximately 180 µM. As this reviewer points out, the date we present clearly demonstrates that Ornidazole does not work more efficiently in cancer cells compared to normal cells, and a JADE2 overexpressing cell line did not have any increased sensitivity to this drug. As such, in our discussion we came to the conclusion that this drug was not suitable for bringing forward. We also discussed the limitations between the methodologies used to assess cellular sensitivity, and further discussed the possibilities that ornidazole could potentially be affecting cancer stem cell fractions as recently demonstrated in melanoma. However, we have chosen to re-write this element of the discussion to further emphasize this and hope that the revised text will meet with the approval of reviewer 2.

The attempt of the authors to assess the effect of the ornidazole is valuable. The expression of JADE2 on the cell line H1819 is not reported (at least in the figure 1).

We thank the reviewers for raising this, and have now included H1819 in the analysis of JADE mRNAs in Figure 1.

Why the authors tested this cell line?

The rationale for including H1819 alongside H1975 was based off the original data that suggested that JADE2 expression was positively associated with EGFR in NSCLC, and whilst no associations were found between JADE2 expression and mutated EGFR, we thought it would be of use to assess the effect of ornidazole on both of these cell lines. In agreement with this it would appear that JADE2 mRNA (at least in cell lines) is fairly ubiquitous (new Figure 1) and not significantly altered between normal bronchial epithelial cell lines and NSCLC cell lines. We hope that this explanation will be sufficient to allay any concerns of the reviewer.

It should be better to test cells expressing or not the JADE2 for the sensitivity to the drug.

We agree with the sentiment underlying this. However, given that the IC50 identified by DepMap was of the order of a dose range of 2.5μM (with a maximum dosage of 10 µM), and the lack of sensitivity seen by us for this agent in all cell lines tested, we are erring on the side of caution here. Moreover, given that ornidazole when tested in an early clinical trial of NSCLC (with a serum/plasma concentration equating to approximately 180 µM), had no overall enhanced effect on NSCLC cancer cells versus normal cells, we do not feel that it warrants additional study for re-purposing as a “stand-alone” agent.

Furthermore, it is not so complicated to test the ornidazole at very low doses with other drugs for treatment of lung cancer to identify, perhaps, a specific effect.

We thank the reviewer for raising this very valid point. We feel that this may be relevant moving forwards, and have added a new series of statements to the conclusions which we feel covers these additional issues, and hope that the reviewer will accept this.

Silencing experiments can better define the role of JADE2. Also, the analysis of expression of the other JADE component of the family can shed some new insight. This reviewer does not know if the other members can substitute JADE2 in some of its functions. 

We thank the reviewer for suggesting these important experiments and have tried to respond as follows:

  1. We agree that silencing experiments would be beneficial, and indeed such experiments are planned for the future. However, we do feel that the current manuscript is publishable without such studies as a biomarker based manuscript.
  2. Regarding the analysis of the expression of other JADE family members, we have indeed added in new analyses of JADE1 & JADE3 not only in NSCLC, but have also provided a pan-cancer analysis which we hope will add to the potential insights of this manuscript.
  3. Possible redundancy issues with regard to additional JADE family proteins affecting/substituting for JADE2. This is an extremely valid point raised, but in the present context we feel that this would be beyond the current manuscripts remit. However, we have in our conclusions section, flagged this important suggestion as a critical area of investigation moving forwards to better define the role(s) of JADE2 (and indeed JADE1 and JADE3). Whilst we understand that this may not be what the reviewer wants, we hope that he/she can accept the revised text and find it acceptable in its present form.

Overall, the manuscript is well written and the effort of the authors is evident. However, the message is weak and the relevance of JADE2 in regulating cell proliferation or other tumor cell features is not clearly demonstrated. In other words, the data shown are not conclusive at all.

We are sorry that the reviewer feels this way. We concede that the relevance of JADE2 in regulating cellular proliferation or otherwise will require substantial future investigations, and have now acknowledged this in the text discussion and conclusions. Nevertheless, we feel that the novel data showing that JADE2 expression is significantly altered in NSCLC and associated with OS warrants publication at the very least as a biomarker study, and hope that the reviewer will accept this so that this invited manuscript submission can now be published.

Round 2

Reviewer 1 Report

The authors have addressed all my concerns.

Minor corrections are required.

Author Response

The authors have addressed all my concerns.

 We thank the reviewer for taking their time to consider our revised manuscript and for finding the manuscript revisions acceptable, and hope that it will be accepted for publication.

Reviewer 2 Report

I understand well the point of view of the authors, but overall I remain on my mind. The authors have added some data, but the relevant experiments are missing again.

Actually, a strong effort has been made in getting results and more importantly in pointing out the relevance of JADE2. However, the manuscript is still preliminary.

English is fine. Please take into account that I am not an expert on this.

Author Response

We thank the reviewer for taking the time to consider our revised manuscript.

I understand well the point of view of the authors, but overall I remain on my mind. The authors have added some data, but the relevant experiments are missing again.

We are sorry that you remain unmoved by our revisions. It may have been better to have rejected it outright at the first instance. We do however, understand your reasoning and indeed in a perfect world in a well-funded laboratory with unlimited access to significant resources such as animal models we would of course have done the experiments suggested by you with respect to animal studies etc. However, such is not the case and we have to carefully consider whether or not such experiments would be worth-while for our available funds. In this instance we had to take the hard decision to close this research project, and would like to publish the existing data as we feel that the scientific community would benefit.

Actually, a strong effort has been made in getting results and more importantly in pointing out the relevance of JADE2. However, the manuscript is still preliminary.

We thank the reviewer for complementing our current efforts to highlight the importance of JADE2 in NSCLC, and the efforts expanded into getting the results presented in the revised manuscript. Whilst we accept that the reviewer still considers the manuscript as being too preliminary, we however feel that the existing data should be made available to the scientific community and consider that the manuscript in its present form remains a publishable item in its own right. If the final decision at this point in time will be to reject this invited manuscript to BioMedicines, we will accept that, and submit it for publication elsewhere.

Round 3

Reviewer 2 Report

Looking again the manuscript, I would endorse it for publication. 

I would add in the paragraph of conclusion that some points relevant to moving forward can be also a limitation of the manuscript.

English language is good.